# An Improved Submerged Mangrove Recognition Index-Based Method for Mapping Mangrove Forests by Removing the Disturbance of Tidal Dynamics and *S. alterniflora*

Qing Xia [1], Ting-Ting He [2,*], Cheng-Zhi Qin [3,4,5], Xue-Min Xing [1] and Wu Xiao [2]

1   Engineering Laboratory of Spatial Information Technology of Highway Geological Disaster Early Warning in Hunan Province, Changsha University of Science and Technology, Changsha 410114, China; xiaqing@csust.edu.cn (Q.X.); xuemin.xing@csust.edu.cn (X.-M.X.)
2   Department of Land Management, Zhejiang University, Hangzhou 310058, China; xiaowu@zju.edu.cn
3   State Key Laboratory of Resources and Environmental Information System, Institute of Geographical Sciences and Natural Resources Research, Chinese Academy of Sciences, Beijing 100101, China; qincz@lreis.ac.cn
4   College of Resources and Environment, University of Chinese Academy of Sciences, Beijing 100049, China
5   Jiangsu Center for Collaborative Innovation in Geographical Information Resource Development and Application, Nanjing 210023, China
*   Correspondence: tthe@zju.edu.cn

**Abstract:** Currently, it is a great challenge for remote sensing technology to accurately map mangrove forests owing to periodic inundation. A submerged mangrove recognition index (SMRI) using two high- and low-tide images was recently proposed to remove the influence of tides and identify mangrove forests. However, when the tidal height of the selected low-tide image is not at the lowest tidal level, the corresponding SMRI does not function well, which results in mangrove forests below the low tidal height being undetected. Furthermore, *Spartina alterniflora Loisel (S. alterniflora)* was introduced to China in 1979 and rapidly spread to become the most serious invasive plant along the Chinese coastline. The current SMRI has failed to distinguish *S. alterniflora* from submerged mangrove forests because of their similar spectral signatures. In this study, an SMRI-based mangrove forest mapping method was developed using the time series of Sentinel-2 images to mitigate the two aforementioned issues. In the proposed method, quantile synthesis was applied to the time series of Sentinel-2 images to generate a lowest-tide synthetic image for creating SMRI to identify submerged mangrove forests. Unsubmerged mangrove forests were classified using a support vector machine, and a preliminary mangrove forest map was created by merging them. In addition, *S. alterniflora* was distinguished from the mangrove forests by analyzing their phenological differences. Finally, mangrove forest mapping was performed by masking *S. alterniflora.* The proposed method was applied to the entire coastline of the Guangxi Province, China. The results showed that it can reliably and accurately identify submerged mangrove forests derived from SMRI by synthesizing low- and high-tide images using quantile synthesis, and the differentiation of *S. alterniflora* using phenological differences results in more accurate mangrove mapping. This work helps to improve the accuracy of mangrove forest mapping using SMRI and its feasibility for coastal wetland monitoring. It also provides data for sustainable management, ecological protection, and restoration of vegetation in coastal zones.

**Keywords:** mangrove forests; vegetation index; Google Earth Engine; Sentinel-2 imagery; mangrove map

## 1. Introduction

Mangrove forests play a critical role in coastal protection and ecosystem functioning worldwide [1–3]. They provide a high diversity of services and a wide variety of goods to coastal communities [4–6]. During the past 50 years, mangrove forests have

experienced rapid losses owing to anthropogenic activities and natural changes to the environment [7,8]. One-third of mangrove forests nearly disappeared globally due to conversion to aquaculture, sea level rise, urban development, overexploitation for timber, and natural disasters [9–11]. Therefore, the rapid and accurate mapping of mangrove forests is necessary for natural resource supervision, protection, and restoration in coastal zones.

Mangrove forests situated in intertidal zones are inaccessible to traditional field-surveying technology. Currently, optical remote sensing and long-term observations are widely used to map coastal mangrove forests [6,12,13]. Numerous classification methods for mangrove forest mapping using remotely sensed imagery have been proposed [3,14–16]. For example, Conchedda et al. [14] proposed an object-based approach using SPOT XS data to distinguish mangrove forests in Low Casamance, Senegal, and achieved an overall accuracy of 86%. Wang et al. [15] applied a clustering-based neural network classifier to the IKONOS data to discriminate mangrove forests in Punta Galeta on the Caribbean coast of Panama. The results showed that the overall accuracy of mangroves was 96%, and the kappa coefficient was 0.88. Although the aforementioned studies achieved mangrove forest mapping, the effect of tides on mangrove forests remains to be determined. Mangrove forests are periodically inundated by tides, especially in regions with low-height mangrove forests and high-fluctuation tides [17]. This periodic inundation results in difficulties for accurately and timely identification of submerged mangrove forests [13,18,19]. To overcome this limitation, it is necessary to obtain remotely sensed images at the lowest tidal heights. Such images are difficult to obtain because the local tidal heights vary continuously, and the time of the local tidal low may not correspond to the time of a satellite transit. Numerous studies have indicated that tides may affect the results of mangrove forest mapping from remotely sensed images [10,20–22]. For example, Zhang et al. [22] proposed a decision tree-based approach using multi-tidal Landsat 5 data and a digital elevation model to map mangroves. The results show that considering the effect of tides on mangrove forests can greatly improve the accuracy of mangrove forest mapping.

In recent years, some mangrove-specific vegetation indices have been proposed to eliminate the influence of tidal dynamics using various remotely sensed images [23]. Tide fluctuation can only submerge seaward, low-height mangrove species, such as the pioneer species *Avicennia marina*, low-height *Aegiceras corniculatum*, or a small part of high-height mangrove species. Numerous indices have been proposed to distinguish mangrove forests in single-tide images. Winarso et al. [24] proposed a mangrove index (MI) using near-infrared (NIR) and shortwave infrared (SWIR) bands from Landsat-8 OLI images (Table 1). However, MI has been applied in one or two case studies, and its feasibility in different regions remains to be verified. Jia et al. [10] proposed a mangrove forest index (MFI) using Sentinel-2 data. MFI is effective for distinguishing submerged mangrove forests utilizing red-edge and NIR bands; however, the applicability of MFI is limited because of the absence of a red-edge band for most remote-sensing sensors. Additionally, a combined mangrove recognition index (CMRI) was proposed by Gupta et al. [25], in which the presence of vegetation was expressed by the normalized difference vegetation index (NDVI), and water information of mangrove forests was expressed by the normalized difference water index (NDWI). CMRI has the same problem as MI. The application of CMRI and MI for mangrove forest mapping should be studied further, especially considering the different sites. However, the common issue here is that these indices have little or no ability to describe tidal variations using a single-tide image.

Some studies have been conducted on mangrove-specific vegetation indices using multi-tide images. Zhang et al. [26] proposed the mangrove recognition index (MRI) using multi-temporal Landsat data with different tidal heights and observed that the MRI provided a sensitive response to changes in wetness and greenness. However, because site-specific vegetation information and land moisture at different tidal heights was not available from remotely sensed images, mangrove forest mapping derived from MRI on a global scale was limited. Xia et al. [27] proposed a submerged mangrove recognition index (SMRI) using high- and low-tide images based on NDVI and NIR bands, which

expressed the presence of vegetation and water, respectively. Compared with the existing mangrove-specific vegetation indices, SMRI is specially designed to distinguish submerged mangrove forests. The calculation of SMRI only involves three bands that are commonly acquired in most sensors, and it does require a large amount of auxiliary information (e.g., field data and moisture information).

**Table 1.** Mangrove-specific vegetation indices (refers to Yang et al. [23]).

| Index Name | Author | Formula | Satellite Image Used |
|---|---|---|---|
| Mangrove recognition index (MRI) | Zhang et al. [26] | $MRI = |GVI_L - GVI_H| \times GVI_L \times (WI_L + WI_H)$ where GVI is the green vegetation index; WI is the wetness index; subscript L indicates low tide; subscript H indicates high tide | Landsat |
| Mangrove index (MI) | Winarso et al. [24] | $MI = (NIR - SWIR/NIR \times SWIR) \times 1000$ | Landsat |
| Normalized difference mangrove index (NDMI) | Shi et al. [28] | $NDMI = (R_{SWIR2} - R_{Green})/(R_{SWIR2} + R_{Green})$ where $R_{SWIR2}$ and $R_{Green}$ are the reflectance values of SWIR2 and green bands, respectively | Landsat |
| Mangrove probability vegetation index (MPVI) | Kumar et al. [29] | $MPVI = \dfrac{n\sum_{i=1}^{n} R_i r_i - \sum_{i=1}^{n} R_i \sum_{i=1}^{n} r_i}{\sqrt{n\sum_{i=1}^{n} R_i^2 - (\sum_{i=1}^{n} R)^2} \sqrt{n\sum_{i=1}^{n} r_i^2 - (\sum_{i=1}^{n} r_i)^2}}$ where n is the total number of bands; $R_i$ is the reflectance value of i band; $r_i$ is the reflectance value of i band for a "candidate spectrum" of mangrove forest | EO-1 Hyperion |
| Combine mangrove recognition index (CMRI) | Gupta et al. [24] | $CMRI = NDVI—NDWI$ where NDWI is the Normalized Difference Water Index. | Landsat |
| Submerged mangrove recognition index (SMRI) | Xia et al. [26] | $SMRI = (NDVI_l - NDVI_h)\cdot(NIR_l - NIR_h)/NIR_h$ where $NDVI_l$—NDVI values at low tide; $NDVI_h$—NDVI values at high tide; $NIR_l$ represents the reflectance values of NIR band at low tide; $NIR_h$ represents the reflectance values of NIR band at high tide. | GaoFen-2 |
| Mangrove forest index (MFI) | Jia et al. [10] | $MFI = [(\rho_{\lambda 1} - \rho_{B\lambda 1}) + (\rho_{\lambda 2} - \rho_{B\lambda 2}) + (\rho_{\lambda 3} - \rho_{B\lambda 3}) + (\rho_{\lambda 4} - \rho_{B\lambda 4})]/4$ where $\rho\lambda$ is the reflectance value of the band center of $\lambda$, and i ranged from 1 to 4; $\lambda 1$, $\lambda 2$, $\lambda 3$, and $\lambda 4$ are the center wavelengths at 705, 740, 783, and 865 nm, respectively. | Sentinel-2 |
| Mangrove vegetation index (MVI) | Baloloy et al. [11] | $MVI = (R_{NIR} - R_{Green})/(R_{SWIR1} - R_{Green})$ where $R_{SWIR1}$ is the reflectance value of SWIR1 band | Sentinel-2/Landsat |
| Normalized intertidal mangrove index (NIMI) | Xu et al. [30] | $NIMI = (3 \times R_4 - (R_6 + R_7 + R_8))/(3 \times R_4 + R_6 + R_7 + R_8)$ where $R_4$, $R_6$, $R_7$, and $R_8$ is the reflectance values of bands 4, 6, 7, and 8 of Sentinel, respectively | Sentinel-2 |
| Optical and synthetic aperture rada (SAR) images combined mangrove index (OSCMI) | Huang et al. [31] | $OSCMI = WI/(NIRB + SWIRB + VV)$ where WI is the sum of NDWI and MNDWI; NIRB is the sum of the reflectance values of Sentinel-2 B6, B7, B8 and B8A; SWIRB is the sum of the reflectance value of Sentinel-2 B11 and B12; VV is the backscatter coefficient of Sentinel-1 VV polarization mode | Sentinel-1/2 |

The SMRI was designed on the basis of two images with different tidal heights. However, two issues must be resolved. The first issue regards the selection of a low-tide image that well approximates the lowest possible tidal height. Tides vary with time, and the variation changes daily. If the tidal height of the selected low-tide image is not a satisfactory approximation to the lowest possible tidal height, mangrove forests inundated under low tidal heights cannot be fully detected. In addition, when multiple images are stitched to map national- or global-scale mangrove forests with SMRI, it is difficult to obtain different

images with the same high and low tidal heights. This would reduce the accuracy of mapping submerged mangrove forests.

The second issue regards the methodology to distinguish *Spartina. alterniflora* from the submerged mangrove forests derived from SMRI. *S. alterniflora* is a perennial salt marsh grass native to North America that was introduced in China in 1979 [32]. It has very strong root systems and prefers to grow with pioneer mangrove species in seaward low-elevation tidal flats [33–35]. Because *S. alterniflora* is characterized by rapid spread and high density, it has become the most predominantly invasive plant along the Chinese coastline [36]. Many *S. alterniflora* species can be found on the fringe of seaward mangrove forests and have a strong disturbance when mapping mangrove forests. SMRI was proposed by analyzing spectral signature differences for submerged vegetation, including that vegetation with chlorophyll absorption characteristics in aquatic environments (i.e., *S. alterniflora*), resulting in poor performance for distinguishing *S. alterniflora* from mangrove forests.

Time series remotely sensed images can provide more tidal variation information for mangrove forest mapping [37–39]. In addition, they can describe the phenological features of *S. alterniflora* and submerged mangrove forests to enhance their spectral separability [40]. There are two prominent phenological periods for *S. alterniflora*: the green and senescence periods. These phenological features were observed across a specific Sentinel-2 image. Visually, the color of mangrove forests is much greener than that of *S. alterniflora* during the senescence period, and this color difference results in their being more distinguishable. The NDVI values of the mangrove forests were much higher than those of *S. alterniflora*. This phenomenon resulted in spectral separation of *S. alterniflora.* Thus, phenological features derived from the time series of Sentinel-2 data were incorporated to eliminate the distribution of *S. alterniflora* in the submerged mangrove forest zone. With the increase in time series data (e.g., Sentinel sensor) and cloud computation resources, some of which are free of charge, such as Google Earth Engine (GEE), a new method based on SMRI using time series images could be developed for effective and accurate mangrove forest mapping.

To mitigate the issues hindering mangrove forest characterization, this study proposes an SMRI-based method for rapid and accurate mapping of mangrove forests using a time series of Sentinel-2 data and GEE. Experiments were conducted to verify the advantages and effectiveness of the proposed method. Specifically, we will (1) apply quantile synthesis to generate high-tide and low-tide synthetic images and discuss the ability of synthetic images to distinguish submerged mangrove forests, (2) utilize phenology-based differences to identify and remove *S. alterniflora* along with discussing the effectiveness of phenology-based differences for removing *S. alterniflora,* and (3) compare the proposed results with those of the existing mangrove forest products.

## 2. Materials and Methods

### 2.1. Study Area

The study area was situated in Guangxi Province (107°58′–109°40′N and 22°00′–21°40′E) (Figure 1). The datum for the Geographic Coordinate System of the China map was WGS-1984. All images and maps were obtained using the WGS-1984 Geographic Coordinate System. The total coastline of Guangxi Province is approximately 1595 km long and the area covered by mangroves per kilometer is the largest in China [41]. The study area has a tropical monsoon-type climate, and the annual average temperature and rainfall are 16 °C–23 °C and 1694 mm, respectively. The major species in the area are *Bruguiera gymnorrhiza, Rhizophora stylosa, Avicennia marina, Aegiceras corniculatum,* and *Kandelia candel.* The tidal type is regularly diurnal, and some mangrove forests are submerged by the sea due to a large tidal range and seaward low-height mangrove forests.

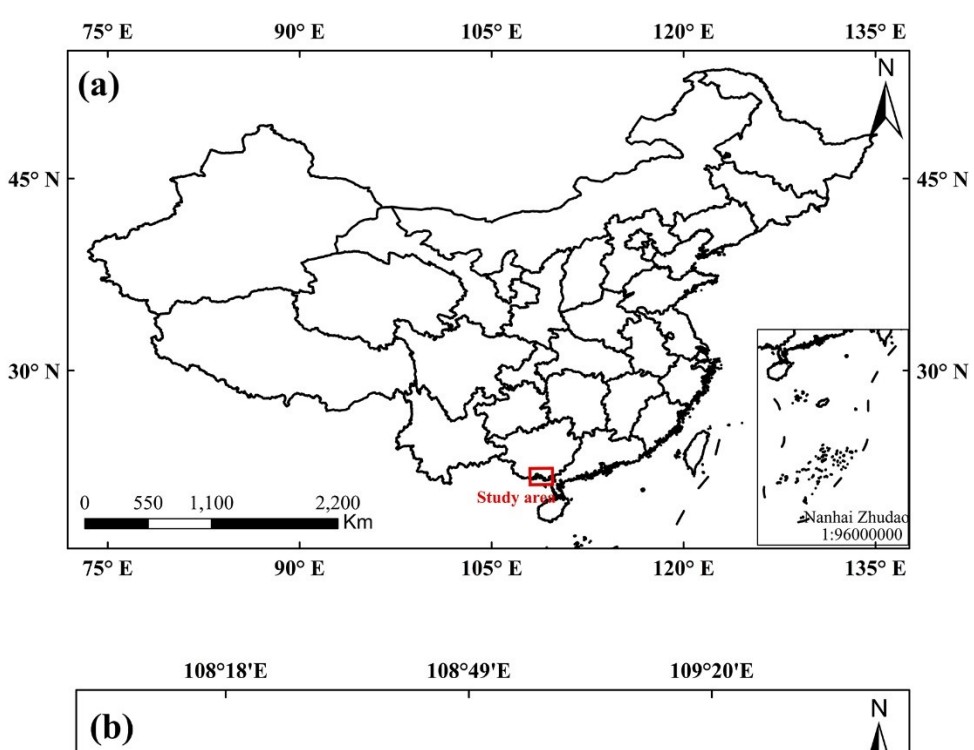

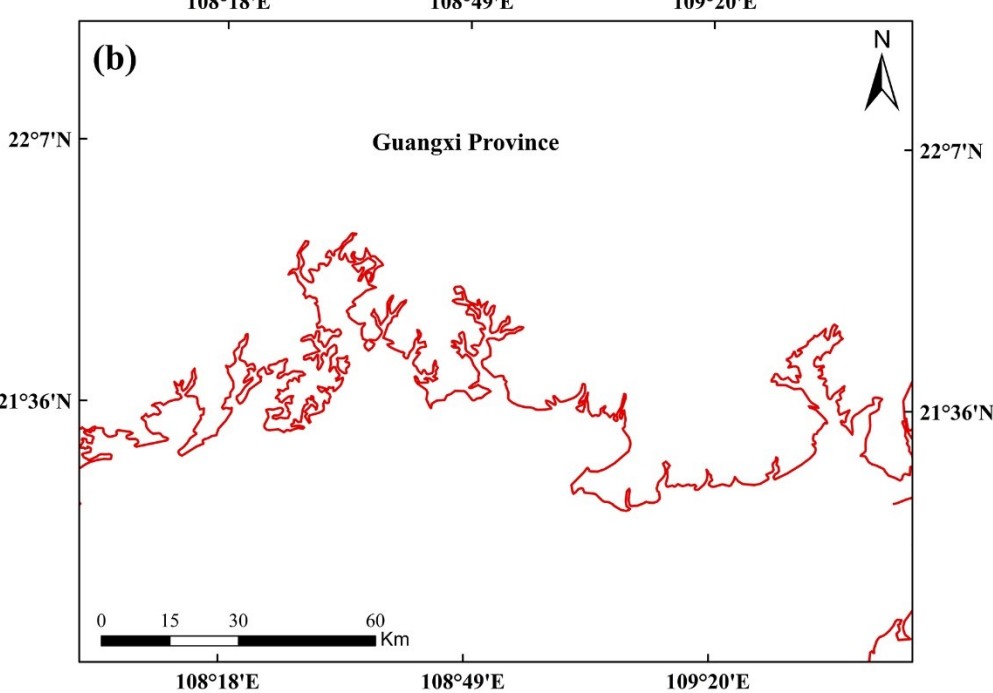

**Figure 1.** Study area; (**a**) China map; (**b**) detailed map of the study area in the Guangxi Province.

### 2.2. Pre-Processing for Sentinel-2 Data

Sentinel-2 multispectral images were downloaded from the European Space Agency (https://scihub.copernicus.eu/dhus/#/home, accessed on 1 January 2019), with three spatial resolutions of 10, 20, and 60 m. Top-of-atmosphere reflectance and surface reflectance data were obtained from Sentinel-2 images. Images with less than 20% cloud cover were selected for analysis. To reduce the effect of clouds, the images were masked using the QA60 band [42]. The surface reflectance data for the Sentinel-2 images with spatial resolutions of 10 m were obtained using GEE. The complete coverage of the surface reflectance data of the study area was 1160 images obtained between January 2019 and December 2020. Through consultation with local experts and government authorities, it was confirmed that the mangrove forests in the study area did not undergo any changes during the observation period. Mangrove forests are periodically inundated by the tide, and the key to

describing tidal status is to identify open surface water. In previous studies, the modified normalized difference water index (mNDWI) has been widely used to identify open surface water [43–45]. The mNDWI value varies with land cover types. The mNDWI of open surface water area will tend toward positive values, whereas other land cover types will be represented by negative values. The Normalized Difference Vegetation Index (NDVI) closely related to green vegetation is a good indicator of vegetation. The NDVI can describe phenological variation for *S. alterniflora* and mangrove forests, and the NDVI values of mangrove forests are higher than those of *S. alterniflora* in the senescence period [36]. Time series for NDVI and mNDWI were used to provide indicative information on phenological variations and tidal inundation.

We surveyed the study area in October 2019 and recorded the location of the sample points (including mangrove forests, non-mangrove forests, S. alterniflora) using a handheld GPS device with a 5 m positional accuracy. The positional accuracy of GPS is accurate enough for the Sentinel-2 images with a spatial resolution of 10 m. Additionally, a high-resolution image from Google Earth was used to collect inaccessible sample points. A total of 1200 sample points comprising mangrove (n = 600, 400 for unsubmerged and 200 for submerged mangrove forests) and non-mangrove forests (n = 600) were collected. A total of 800 (400 for mangrove and 400 for non-mangrove forests) and 400 (200 for mangrove and 200 for non-mangrove forests) sample points were used for training and validation, respectively (Table 2, Figure 2).

**Table 2.** Number and type of sample points in the study area.

| | | Types | Number | Total Number |
|---|---|---|---|---|
| Training and validating | Mangroves | Submerged | 150 | 600 |
| | | Non-submerged | 450 | |
| | Non-mangroves | *S. alterniflora* | 90 | 600 |
| | | Tidal flats | 80 | |
| | | Water | 200 | |
| | | Offshore ponds | 150 | |
| | | Built-up land | 80 | |
| Training | Mangroves | Submerged | 100 | 400 |
| | | Non-submerged | 300 | |
| | Non-mangroves | | 400 | 400 |
| Validating | Mangroves | Submerged | 50 | 200 |
| | | Non-submerged | 150 | |
| | Non-mangroves | | 200 | 200 |

### 2.3. SMRI-Based Method for Mangrove Forests Mapping

First, we visually interpreted the coastal boundary zone covering the mangrove forests. Second, we generated high-tide and low-tide synthetic images using quantile synthesis based on the time series of Sentinel-2 data and the GEE platform. Quantile synthesis uses quantiles at each pixel of the time-series images to estimate the tidal datum from the pixel. Here, the low and high tidal data were characterized using the 10th and 90th quantile, respectively. The 10th quantile was determined to describe the low tidal datum due to the removal of poor-quality images affected by clouds and shadows. The 90th quantile is sufficient to represent high tidal data based on our visual interpretation, without an assumption about the transit time of satellites. Subsequently, the submerged mangrove forest areas were distinguished using SMRI. Non-submerged mangrove forests were classified using a support vector machine (SVM) classifier, and a preliminary mangrove forests map was obtained by merging them. Finally, *S. alterniflora* was distinguished by analyzing phenological differences, and a final mangrove forest map was created by masking *S. alterniflora* (Figure 3).

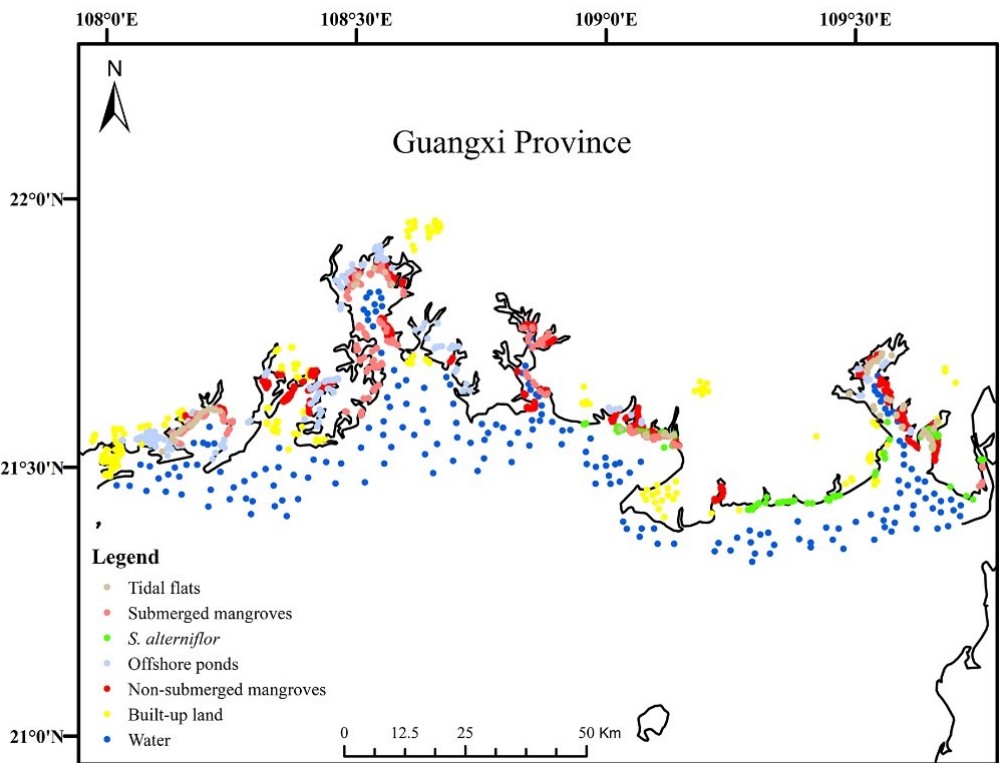

**Figure 2.** Location map of sample points derived from seven land cover types.

### 2.3.1. Coastal Boundary Zone

Mangrove forests typically grow in intertidal areas. We digitized the entire Guangxi Province's coastal boundary zone in GEE by visually interpreting Sentinel-2 data from January 2019 to December 2020. To ensure that all scattered patches of mangrove forests were included, a 20 km radius buffer was generated along the boundary.

### 2.3.2. Generation of Low-Tide of and High-Tide Synthetic Images

The mNDWI image time series was obtained from the time series of Sentinel-2 images (Figure 4). Following the method reported for generating tidal data in mangrove forest mapping in China [13], we counted the mNDWI values at each pixel from the mNDWI images to form a histogram and found the 10th and 90th quantiles at each pixel of the mNDWI images (from January 2019 to December 2020). We subsequently calculated the average mNDWI values at the 10th and 90th quantiles as the low- and high-tide quantile synthesis results, respectively. The 10th quantile was selected as a threshold because it is difficult for mangrove forests to be found in low-tide synthetic images with less than the 10th quantile based on visual interpretation. Finally, submerged mangrove forests were mapped with the criteria (SMRI > 0) based on the acquired synthesis of low-tide and high-tide images, and the SMRI equation is as follows:

$$SMRI = (NDVI_l - NDVI_h) \cdot \frac{NIR_l - NIR_h}{NIR_h} \tag{1}$$

where $NDVI_l$ is the NDVI value at low tide, $NDVI_h$ is the NDVI value at high tide, and $NIR_l$ and $NIR_h$ are the reflectance values of the NIR band at low and high tide, respectively.

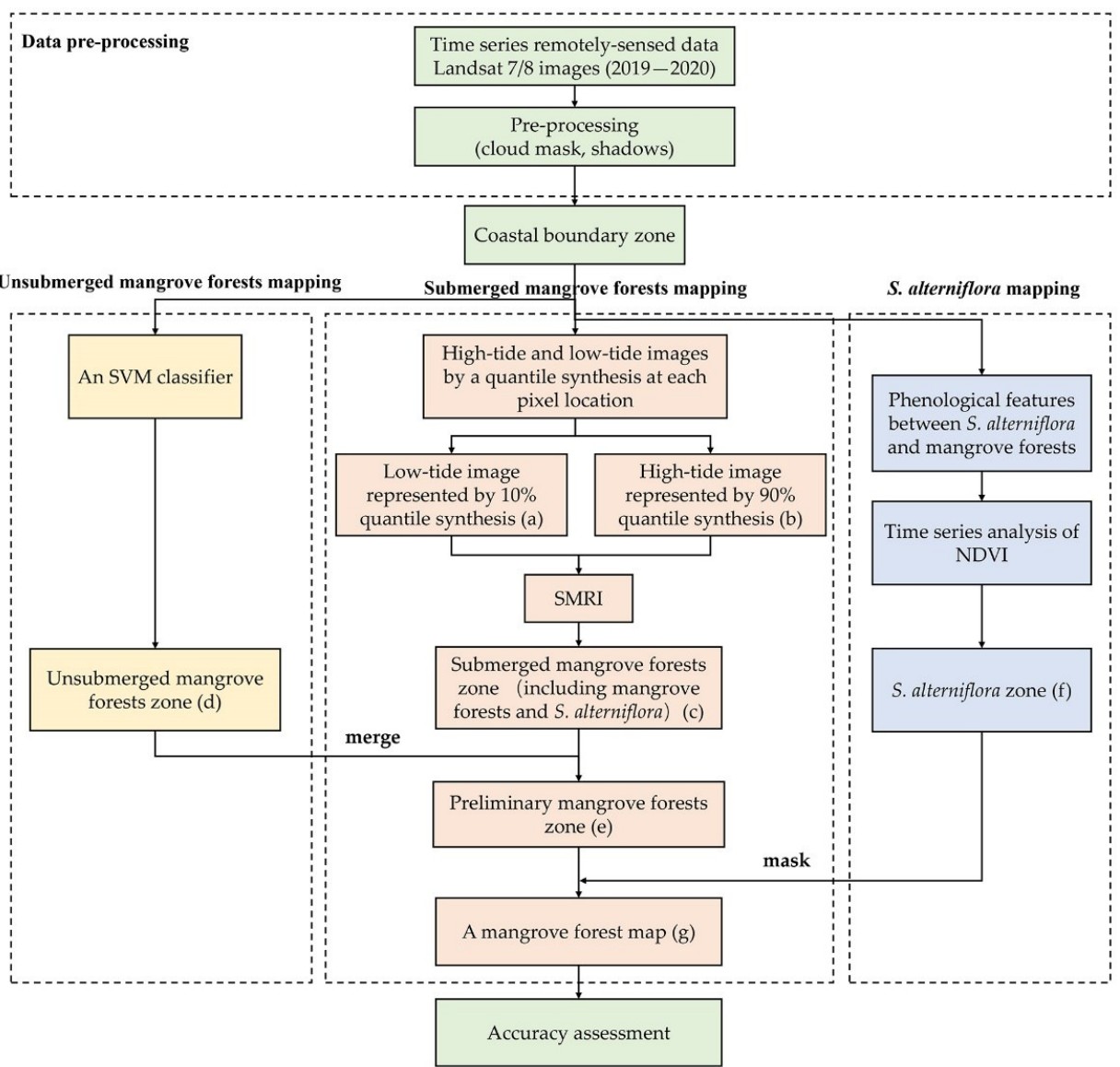

**Figure 3.** Overall study workflow.

### 2.3.3. Phenology-Based *S. alterniflora* Mapping

Submerged mangrove forest areas obtained from SMRI include underwater mangrove forests and *S. alterniflora* because of their similar spectral signatures. As there are phenological differences between *S. alterniflora* and mangrove forests, they provide a good opportunity to separate them with spectral separability [46–49]. The NDVI is a widely used indicator of mangrove forest classification [35]. Thus, we used NDVI as a key phenological indicator to separate *S. alterniflora* from the submerged mangrove forest zones.

### 2.3.4. SMRI-Based Mangrove Forests Mapping Method

After generating high- and low-tide synthetic images, the SMRI was calculated, and the SMRI index calculation resulted in a grayscale image. Otsu's automatic thresholding method was used to generate binary images for submerged mangrove forests (with values of 1) and other land cover types (with values of 0) [50] (Figure 5a–c). The Otsu thresholding method is a non-parametric approach, and this algorithm assumed that the grayscale image contains two classes of pixels. The algorithm subsequently calculated the optimum threshold separating the two classes so that their intra-class variance was minimal, or their inter-class variance was maximal [51]. We then used an SVM classifier to map the unsubmerged mangrove forests using the same sample points (Figure 5d). SVM is a fast

and efficient machine-learning technique that is well adapted for solving non-linear, high dimensional space classifications [52]. SVM utilizes a user-defined kernel function to define a set of non-linear decision boundaries in the original dataset as linear boundaries of a higher-dimensional construct. SVMs attempt to determine the optimal separating hyperplane through an optimization approach utilizing Lagrange multipliers and quadratic programming methods [53]. We used the radial basis function kernel to implement the SVM algorithm because it is commonly used and exhibits good performance [54,55]. Two parameters must be parameterized when applying the SVM classifier with the radial basis function kernel: a regularization constant $C$ and a kernel hyperparameter $\gamma$. The grid search algorithm [50] was used to tune these parameters to the optimum values, that is, $C = 100$ and $\gamma = 0.059$ in this study. A preliminary mangrove forest zone was mapped by merging the unsubmerged and submerged mangrove forest zones (Figure 5e). Finally, a mangrove forest zone was mapped by masking the *S. alterniflora* zone from the preliminary mangrove forest zone (Figure 5f,g).

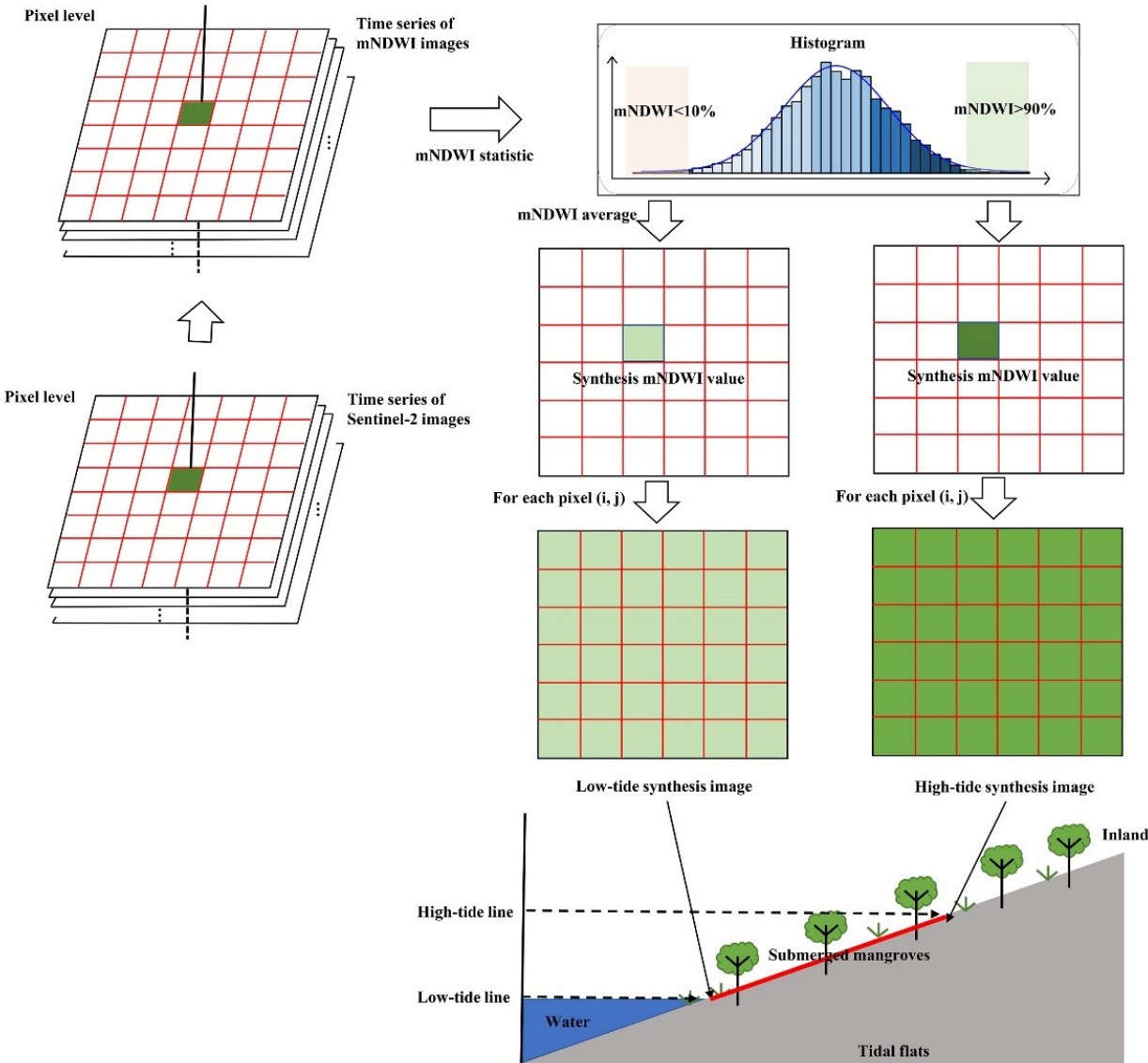

**Figure 4.** Basic concept for generating high- and low-tide synthesis images.

To evaluate performance, an accuracy assessment was performed using sample points from the field investigation. SVM-based classification requires training sets (800 sample points: 400 for mangrove and 400 for non-mangrove forests) as reference signatures that

are used to classify the whole population of pixels and validation sets (400 sample points: 200 for mangrove and 200 for non-mangrove forests) for verification.

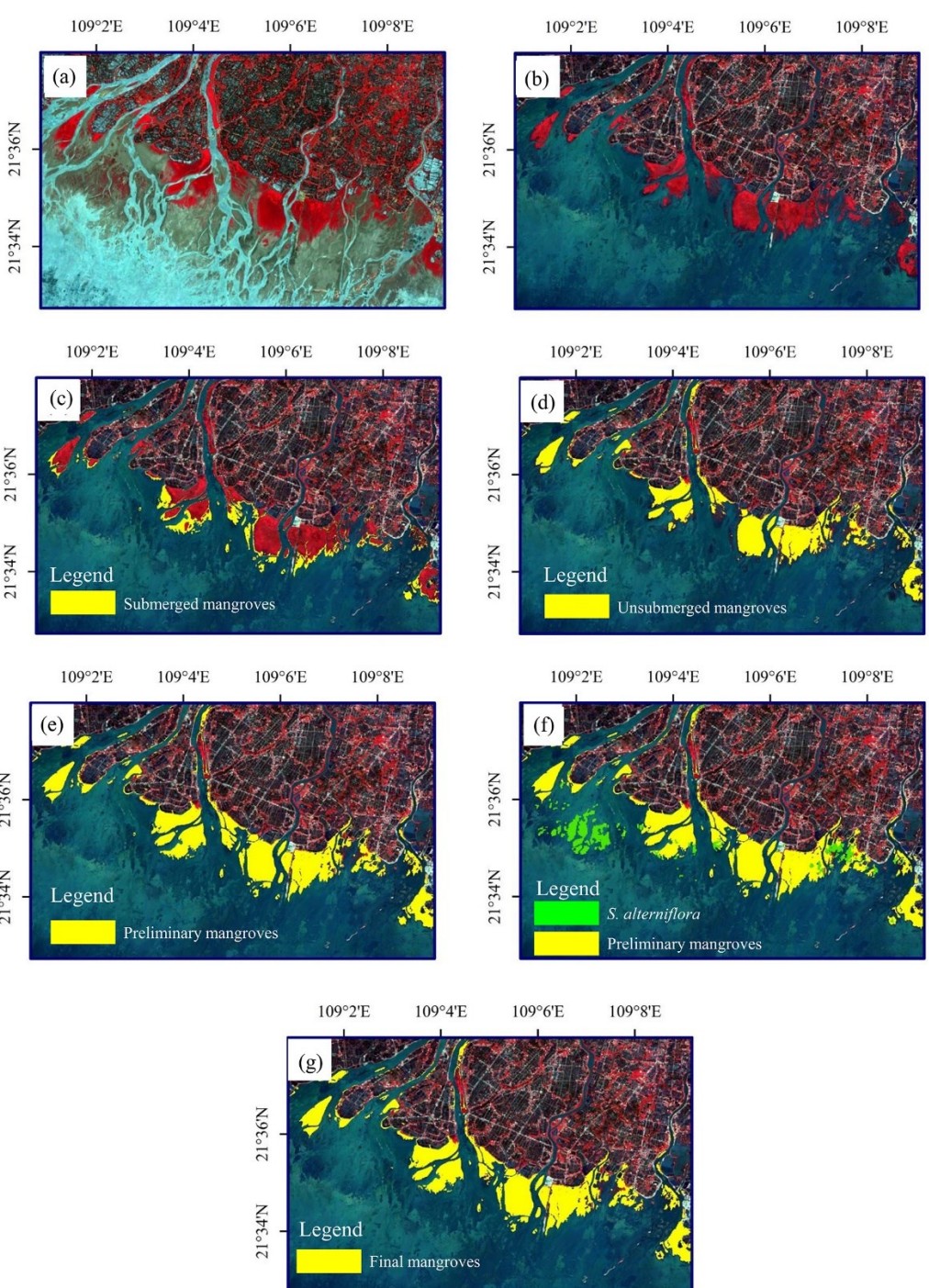

**Figure 5.** Selected case study flowchart: (**a**) Synthetic low-tide Sentinel-2 image, (**b**) synthetic high-tide Sentinel-2 image, (**c**) submerged mangrove forests derived from submerged mangrove recognition index (SMRI), (**d**) unsubmerged mangrove forests derived from support vector machine (SVM), (**e**) preliminary mangrove forest map, (**f**) *S. alterniflora* map, and (**g**) final mangrove forests map.

## 3. Results

A total of 1000 sample points for *S. alterniflora* (n = 500) and mangrove forests (n = 500 including submerged and non-submerged mangrove forests) were selected from the field investigation and Google Earth. To capture the ideal annual phenological features of

*S. alterniflora* and mangrove forests, we generated an average annual NDVI profile to explore phenological differences using Sentinel-2 time series data (Figure 6). Two critical phenological periods (green and senescence) for *S. alterniflora* were identified and separated by analyzing the NDVI temporal profiles.

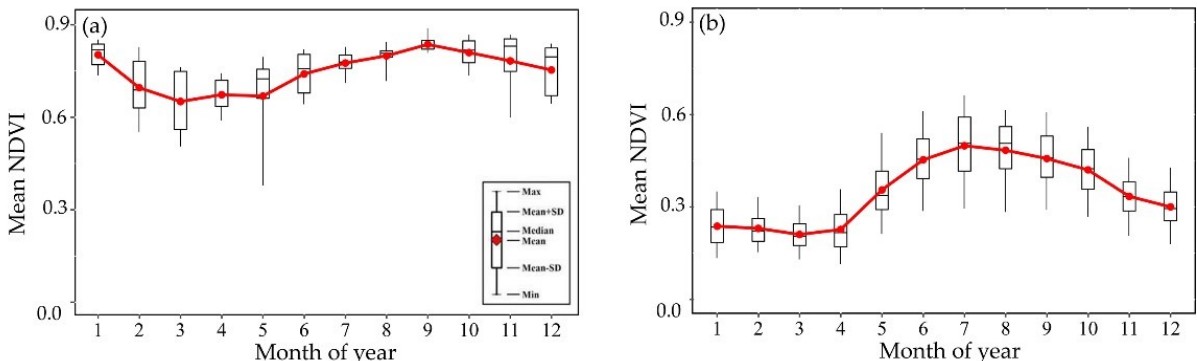

**Figure 6.** NDVI temporal profile: (**a**) mangrove forests and, (**b**) *S. alterniflora*.

The mean NDVI values of mangrove forests were greater than 0.6 for the entire year (Figure 4). Before April, the mean NDVI values of *S. alterniflora* were all less than 0.35, whereas the values between June and October were much greater than 0.35. Thus, there is a clear boundary separating the phenological periods of *S. alterniflora,* that is, the senescence period from January to April and the green period from June to October. We used the mean NDVI value < 0.35 from January to April to avoid the influence of *S. alterniflora*.

For comparison, an SVM classifier was used to generate a mangrove forest map. Figure 7 illustrates the mapping results of the two methods, including the proposed method (SMRI+ Quantile Synthetic images, SMRI + QS) and SVM classifier method (SVM). Based on the SMRI + QS mapping results, submerged mangrove forests were effectively classified as mangrove forests using the SMRI. Mangrove forests are mainly distributed in the Beilun Estuary National Nature Reserve (BNNR), Shankou Mangrove Nature Reserve (SNNR), and Maoweihai Mangrove Provincial Nature Reserve (MPNR). Some scattered patches of mangrove forests are distributed along the coastline and estuaries. From the SVM mapping results, mangrove forests above water were not identified. The total mangrove forest areas from SMRI + QS and SVM were 9110.17 ha and 7616.94 ha, respectively. The area of submerged mangrove forests using SMRI + QS was 1493.23 ha greater than that obtained using SVM. This indicates that most submerged mangrove forests were accurately distinguished, which was attributed to the generation of high- and low-tide synthetic images using quantile synthesis and the separation of *S. alterniflora* from submerged mangrove forests using spectral separability.

With the increasing availability of open data (Sentinel-2) and open-source programs (GEE), we successfully applied the proposed method to the study area and achieved satisfactory accuracy (Table 3). The overall accuracies were 90.5% for SVM and 93.8% for SMRI + QS. The kappa coefficient of 0.87 for SMRI + QS was greater than that of 0.81 for SVM. The producer and user accuracies of mangrove forests were 91.5% and 89.7% for SVM, and 94.5% and 93.1% for SMRI + QS, respectively. The proposed method increased the accuracy of mangrove forest mapping derived from improved SMRI.

**Table 3.** Accuracy assessment for the study area.

| Method | Class | Reference | | Producer Accuracy | User Accuracy | Overall Accuracy | Kappa | Area (ha) |
|---|---|---|---|---|---|---|---|---|
| | | Mangroves | Non-Mangroves | | | | | |
| SVM | Mangroves | 183 | 17 | 91.5% | 89.7% | 90.5% | 0.81 | 7616.94 |
| | Non-mangroves | 21 | 179 | 89.5% | 91.3% | | | |
| SMRI + QS | Mangroves | 189 | 11 | 94.5% | 93.1% | 93.8% | 0.87 | 9110.17 |
| | Non-mangroves | 14 | 186 | 93.0% | 94.4% | | | |

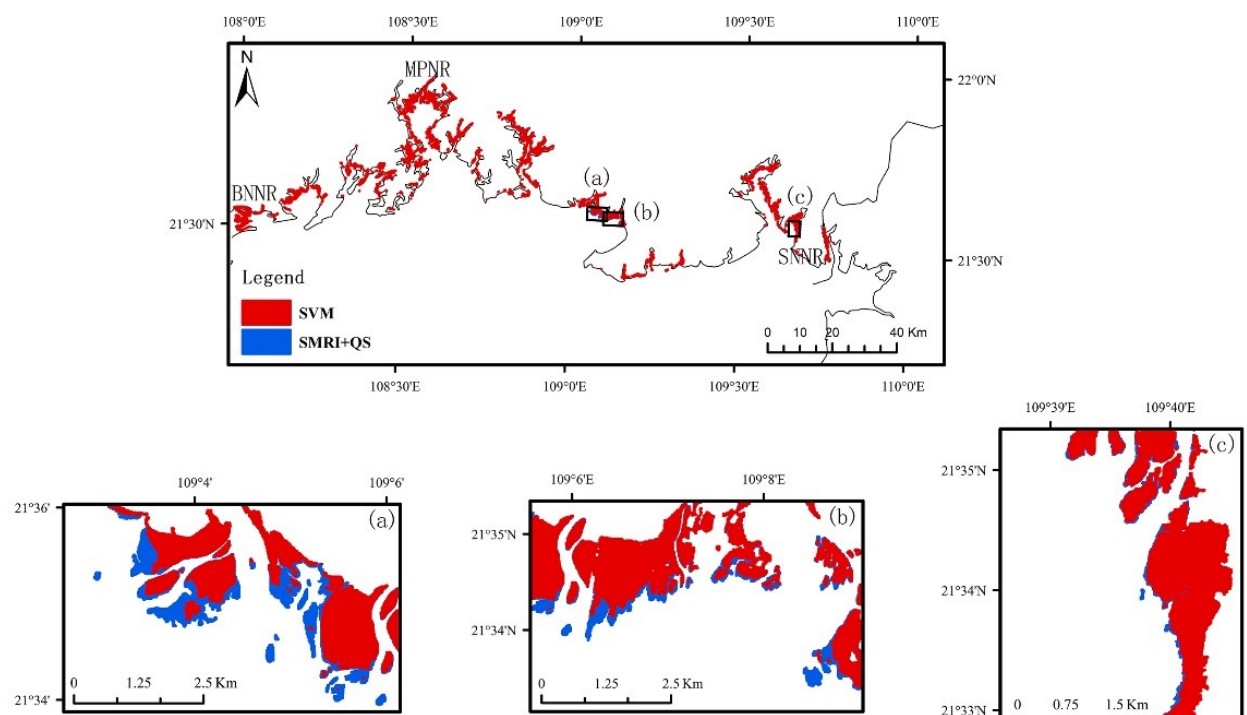

**Figure 7.** Mapping results of mangrove forests from SMRI + QS (red plus blue) and SVM (red). BNNR, Beilun Estuary National Nature Reserve; MPNR, Maoweihai Mangrove Provincial Nature Reserve; SNNR, Shankou Mangrove Nature Reserve. (**a–c**) are the detailed maps.

To validate the advantages of quantile synthesis, we selected the highest- and lowest-tide images from the time series of Sentinel-2 images (2019.01–2020.12) and achieved two mangrove forest mapping results using quantile synthetic (SMRI + QS) images and without using quantile synthetic (SMRI + HL) images (Figure 8). SMRI + HL refers to the mangrove forest mapping results with two randomly selected high- and low-tide images. Along the entire Guangxi Province coastline, we selected three sub-study areas for comparison.

To quantitatively discuss the effect of *S. alterniflora* on mangrove forest mapping, the initial mangrove forest results (before removing *S. alterniflora* using the proposed method, called initial SMRI + QS), the final mangrove results (after removing *S. alterniflora*, called SMRI + QS), and *S. alterniflora* results are mapped in Figure 9.

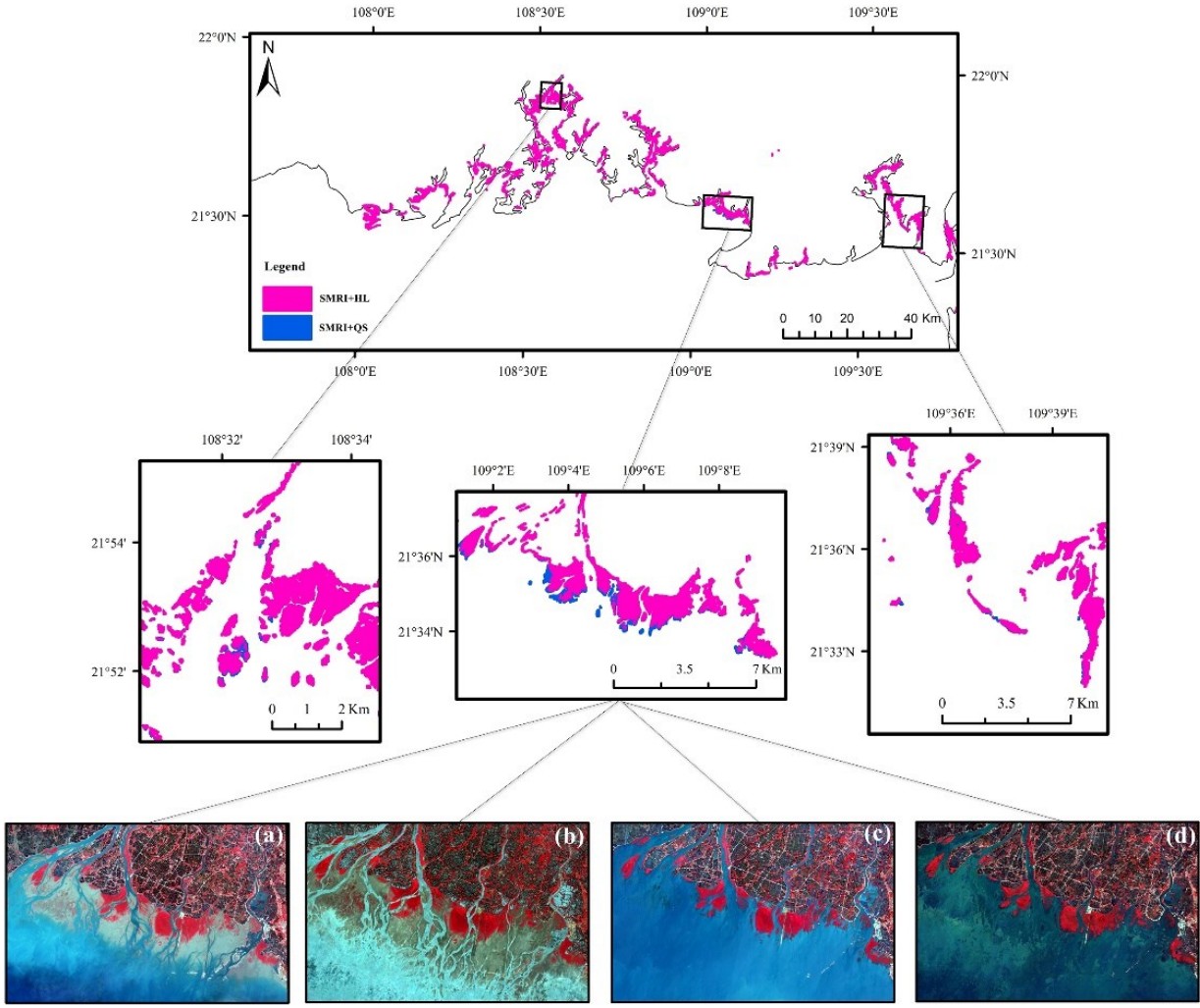

**Figure 8.** Distribution of mangrove forests derived with SMRI using quantile synthesis images and selected high- and low-tide images: (**a**) low-tide Sentinel-2 image, (**b**) synthesis low-tide Sentinel-2 images, (**c**) high-tide Sentinel-2 image, and (**d**) synthesis high-tide Sentinel-2 image.

Mangrove forest maps have already been published for our study area. Therefore, the mangrove forest map generated in this study was compared with the following public data: (1) The Chinese Academy of Sciences Mangroves (CASM), Jia et al. [12]; (2) The Mangrove Forest Distribution Map (MFDM), Chen et al. [20]; and (3) The Global Mangrove Map (GMW), Murray et al. [56]. These maps were generated using images from 2015, 2015, and 2019 for the CASM, MFDM, and GMW, respectively. A map of the four mangrove products is shown in Figure 10.

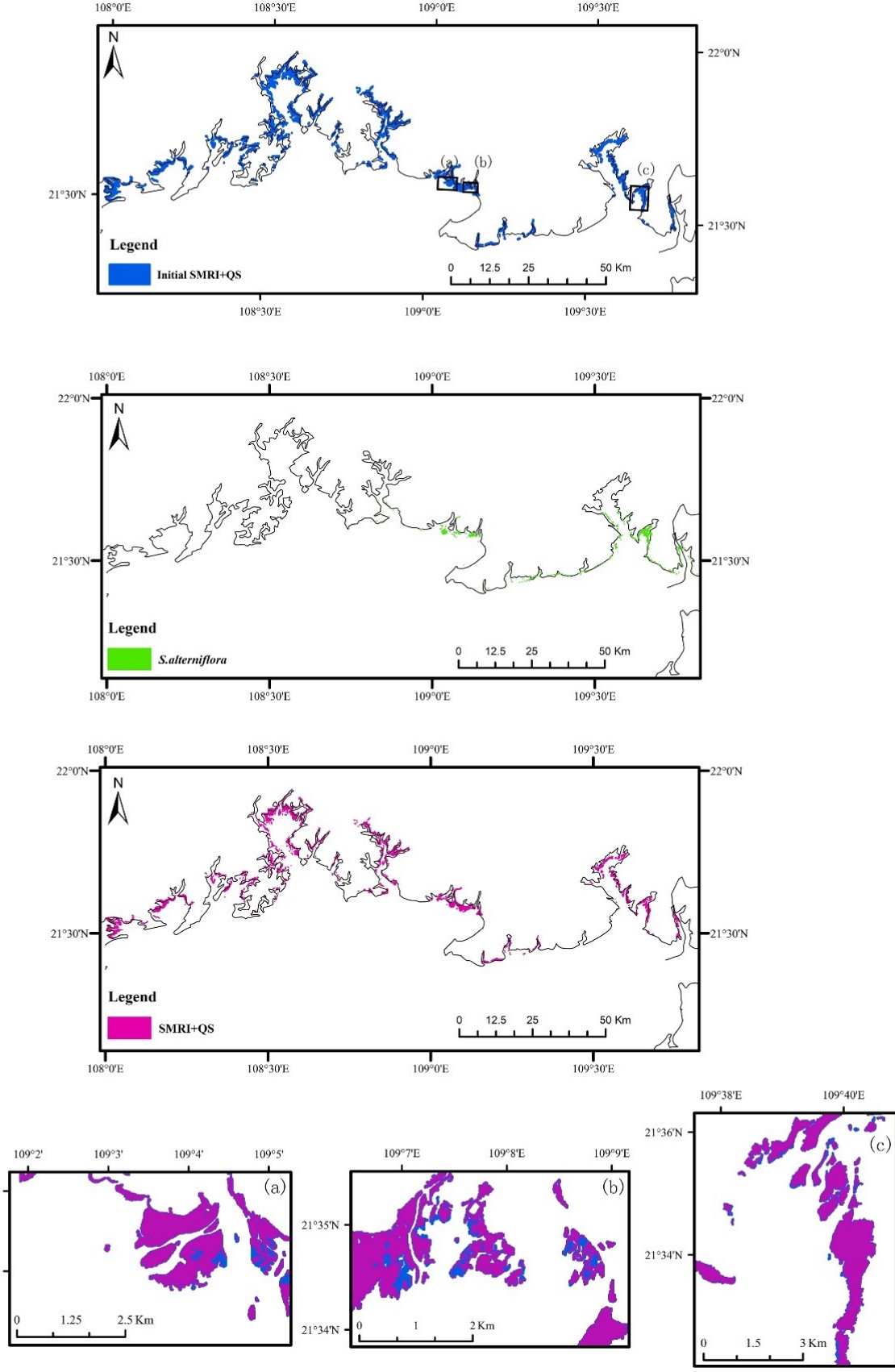

**Figure 9.** Distribution of mangrove forests derived from SMRI using quantile synthesis and *S. alterniflora*. (**a–c**) present detailed results from initial SMRI + QS and SMRI + QS.

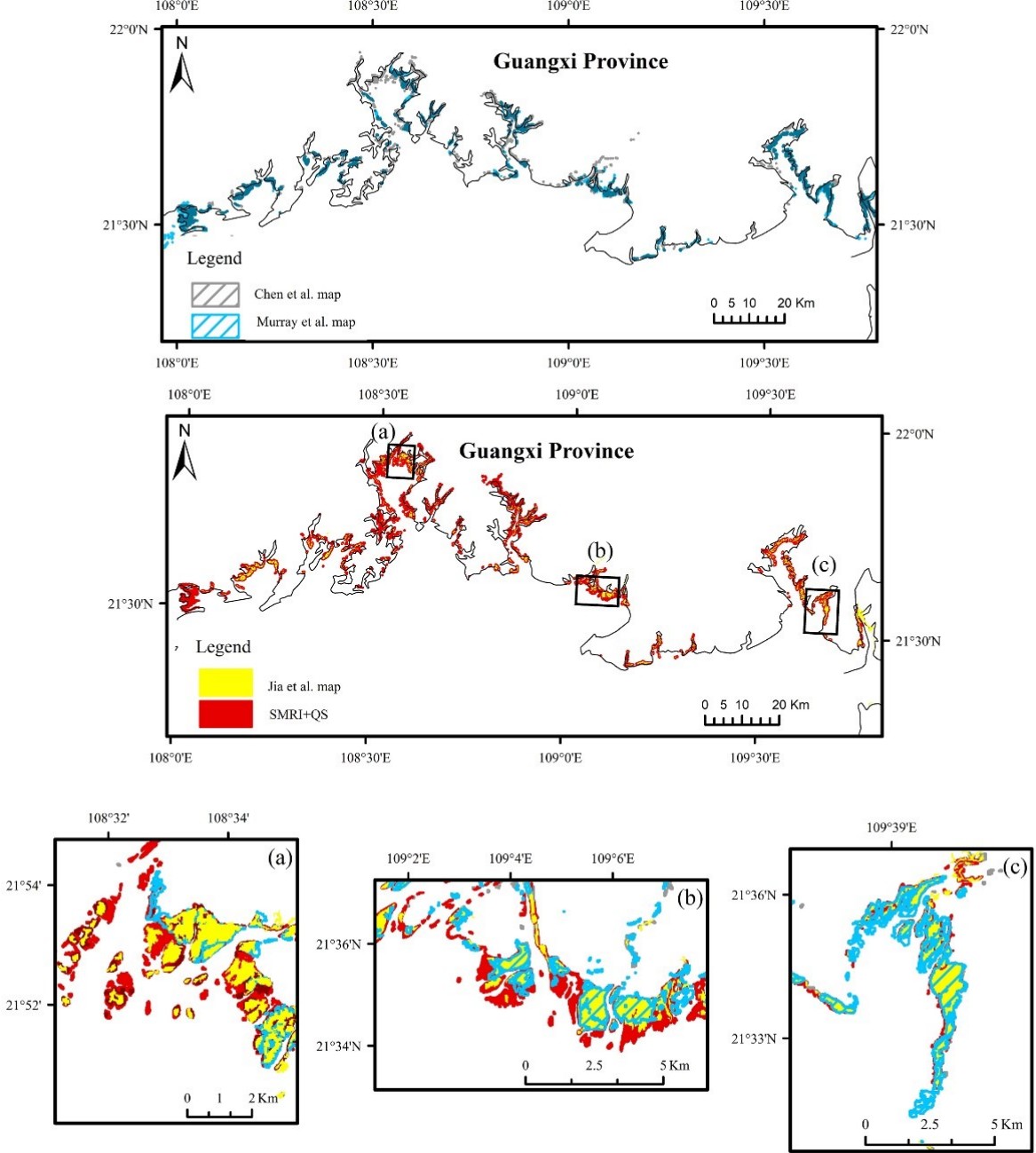

**Figure 10.** Comparison of four mangrove forest products ((**a**–**c**) present the detailed sub-map from four results).

## 4. Discussion

### 4.1. Generation of Low-Tide and High-Tide Synthesis Images

Before identifying the submerged mangrove forest zone using SMRI, we tried to obtain the lowest- and highest-tide images calculated by quantile synthesis using the time series of mNDWI images (see Section 2.3.2). Subsequently, the SMRI index was calculated in each pixel, and the calculated SMRI result was displayed in the form of a grayscale image. Tidal dynamics are an important factor affecting the distribution of submerged mangrove forests derived from SMRI. The sentinel sensor only captures one moment of the entire tidal range at a fixed location, and it is difficult to capture the moment at extremely low and high tides [57]. Using the time series of Sentinel-2 images during a given period, it is possible to capture high- and low-tide images.

From the SMRI + QS results (Figure 8), the synthetic high- and low-tide images can be maximized close to the highest and lowest tidal heights, respectively. In addition, the largest tidal difference can be obtained for more submerged mangrove forests. From the results of SMRI + HL, mangrove forests under the low tidal height from the selected low-tide image could not be detected, resulting in the loss of mangrove area. Quantile synthesis results in a more effective separation of high- and low-tide images, and the submerged mangrove forests derived from SMRI are more accurate. Thus, this is an advantage of the proposed method when determining high-tide and low-tide images using SMRI to map submerged mangrove forests. Furthermore, the generation of high- and low-tide synthetic images using the quantile synthesis is more automatic and reliable because it leads to less uncertainty for periodic inundation and overcomes the limitations of poor observations or clouds.

In addition, one may obtain more images with different tidal heights using quantile synthesis of satellite observations based on different application purposes. Zhao et al. [13] tested the applicability of quantile synthesis using synthetic aperture radar (SAR) images and extended the application of quantile synthesis to optical images (i.e., Sentinel sensor) and verified its applicability.

### 4.2. Separation of S. alterniflora

Phenological features are important for distinguishing *S. alterniflora* from mangrove forests. Mangrove forests are marginally affected by seasonal changes, but *S. alterniflora* is always influenced by seasonal changes. From the results in Figure 9, many *S. alterniflora* patches were misclassified as submerged mangrove forests, resulting in the overestimation of mangrove forest area. The separation of *S. alterniflora* using phenological differences results in a more accurate mangrove forest mapping. However, it is extremely difficult for some mixed pixels of *S. alterniflora* to be distinguished from submerged mangrove forests because of the limited spatial resolution of the Sentinel-2 sensor [35,58].

Additionally, we only employed two phenological periods in this study rather than stacking numerous phenological periods to reduce the demand for data. Theoretically, most of the effective phenological information is accommodated by these two specific periods and provides additional phenological information for *S. alterniflora* separation.

The areas of the initial SMRI + QS, SMRI + QS and *S. alterniflora* were calculated. The total area of initial SMRI + QS prior to removing *S. alterniflora* was 9308.86 ha, and the area of SMRI + QS after removing *S. alterniflora* was 9110.17 ha. The total area covered by *S. alterniflora* was 1088.08 ha. The area prior to removing *S. alterniflora* was 198.69 ha larger than that after removing *S. alterniflora*.

### 4.3. Comparison with Other Mangrove Forests Mapping Products

From the results in Figure 10, the differences between this map and other mangrove forest products can be explained. Jia et al. [12] neglected tidal inundation by applying single-phase Landsat data to map mangrove forests. Chen et al. [20] used the relationship between specific mangrove indices and annual mean NDVI to map mangrove forests. The study by Chen et al. was implicit in synthesized quantiles. Murray et al. [56] created a global mangrove forest map using random forest classification with 30 m resolution Landsat data. Comparing the map derived from this study with those from the three mangrove forest products, the first advantage of this study is the removal of the *S. alterniflora* influence. *S. alterniflora* near seaward mangrove forests flourish and have a significant impact on mangrove forest mapping. If the effect of *S. alterniflora* is neglected, the area of mangrove forests can be overestimated. The second advantage of this study is that it distinguishes submerged mangrove forests. If tidal inundation was considered, the mangrove forest area would be underestimated. We also calculated the area of the three products, and the areas of mangrove forest mapping were 6621, 6849, and 6192 ha for the CASM, MFDM, and GMW, respectively. The mangrove forest area in this study (9110.17 ha) was larger than that obtained using the other products. The 10 m resolution from Sentinel-2 data contributes to

finer mangrove forest identification, and more small, fragmented patches near landward ponds or along a tidal river can be distinguished. Moreover, this map in Guangxi Province, Chin, is the first mangrove forest map accessible to the public.

### 4.4. Limitations for the Proposed Method

The classification results presented the so-called "salt-and-pepper" level of characterization because the proposed method is conducted at the pixel level. In our future work, object-based classification will be conducted to eliminate the "salt-and-pepper" effect of the proposed method. Additionally, optical images are susceptible to clouds. For areas with a rainy season in phenology, few or even no optical images are available for one or two months because of cloud cover. This results in phenological feature losses for one or two months of a year but does not affect the annual phenological features. The potential of combining multi-source data (i.e., Landsat, Sentinel, and SAR data) to compensate for this limitation with the proposed method merits future investigation.

### 5. Conclusions

In this study, an improved SMRI-based method for mangrove forest mapping was developed using a time series of Sentinel-2 images and the GEE platform. The proposed method generated high- and low-tide synthetic images using quantile synthesis of satellite observations rather than selecting two images from available images, which removed the disturbance of tidal dynamics during image acquisition. This resulted in more effective and reliable determination of high- and low-tide images; consequently, the submerged mangrove forests derived from SMRI were more accurate. In addition, the proposed method removed the disturbance caused by *S. alterniflora* from the submerged mangrove forest zone based on spectral separability. It can maximize the use of available images and is more effective and accurate for mangrove forest mapping, especially in areas affected by tides. The proposed method removed the disturbance of tidal dynamics and *S. alterniflora* invasion and can accurately map mangrove forests. This study provides a mangrove forest classification method that has the potential to be applied to coastal monitoring.

**Author Contributions:** Q.X.: conceptualization, methodology, writing—original draft, review, editing, and supervision. T.-T.H.: Software, validation, investigation. C.-Z.Q.: conceptualization and methodology. X.-M.X.: data curation and analysis. W.X.: language editing. All authors have read and agreed to the published version of the manuscript.

**Funding:** This research work was funded by National Natural Science Foundation of China (No. 42101356, 42074033). The work was also supported by Natural Science Foundation of Hunan Province, China (No. 2022JJ40473) and the Research Foundation of Education Bureau of Hunan Province, China (No. 19C0042). Cheng-Zhi Qin thanks the support from the Science and Technology Basic Resources Investigation Program of China (No. 2017FY100706). The authors are grateful to all members of research group for their help in data process.

**Data Availability Statement:** All data are available upon request from the corresponding author.

**Conflicts of Interest:** The authors declare that they have no known competing financial interest or personal relationships that could have appeared to influence the work reported in this paper.

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
