# Peer review of "An Improved Submerged Mangrove Recognition Index-Based Method for Mapping Mangrove Forests by Removing the Disturbance of Tidal Dynamics and S. alterniflora"

_remotesensing, doi:10.3390/rs14133112_

Round 1

Reviewer 1 Report

The authors try to address an interesting issue in the field of mangroves. However, the following things should be improved:

  • the effect of tidal inundation and different species
  • Seasonal differences of mangroves and their effect on different indices used
  • structure of the paper especially results and discussion; the discussion should be improved a lot
  • Introduction should be improved with a clear motivation to the study and clear objectives

Please see the attached document for more comments.

Author Response

Reviewer #1:

The authors try to address an interesting issue in the field of mangroves. However, the following things should be improved

Please see the attached document for more comments.

  1. [Comment]: Existing mangrove indices include not only 5 species listed in Table 1, but also MVI, NDMI, NIMI, OSCMI, etc.

[Response]: We appreciate the comment. Following your suggestion, we have added more mangrove indices including NDMI, MPVI, MVI, NIMI and OSCMI in Table 1 in the revised manuscript.

  1. [Comment]: Why did this study choose 2019-2020 images instead of one year?

[Response]: Thanks for this comment. Available images with less than 20% cloud cover in one year are not enough to generate low-tidal and high-tidal synthetic images. Thus, we extend the time span to two years.

  1. [Comment]: The type and number of sample sites in this paper are not clearly expressed: the non-mangrove category should include spartina alterniflora; The number of submerged and unsubmerged mangrove samples in training samples and verification samples is not explained separately, but only the total number of mangrove samples is provided. Distribution map of sample points and quantity table can be provided. 

[Response]: We appreciate this comment and recognize that more detailed information is needed. The additional detail of samples has been provided in Table 2. We also added Fig. 2 to describe the location of samples in Line 201.

  1. [Comment]: Why choose 20 km as buffer radius?

[Response]: We appreciate this comment. We digitized the whole Guangxi Province’s coastal boundary zone in GEE by visually interpreting Sentinel-2 images. During the visually interpretation, we found that the farthest accessible boundary of seawards or landward mangroves did not exceed 20 km from the coastline. Thus, to make sure that all scattered patches of mangrove forests were included, a 20 km radius buffer was generated along the boundary.

  1. [Comment]: Quantile synthesis of high tide and low tide images: 10% and 90% in FIG.2 correspond to high tide and low tide respectively, while 10% and 90% in Section 2.3.2 correspond to low tide and high tide respectively. Which one is correct? Also, why 10 percent, 90 percent instead of 5 percent, 95 percent, or whatever?

[Response]: Sorry for this mistake. 10% and 90% in Fig.2 should correspond to low-tide and high-tide images. We have fixed it in Fig. 2.

We determine 10% and 90% as quantile for low-tide and high-tide synthetic images. There are two reasons as follows: 1) We try to select Sentinel-2 images with less than 20% cloud cover in the image preprocessing stage, however, it is impossible to remove poor-quality images affected by clouds and shadows. Thus, it is not reasonable to take the maximum and minimum at each pixel of mNDWI images as the highest tidal height and lowest tidal height. To remove the disturbance of poor-quality images, we considered quantile synthesis in this study. 2) Based on our visual interpretation, mangrove forests growing in the low-tide synthetic images with quantile less than 10th are difficult to be found. Thus, we selected 10th quantile as the low-tide synthetic image. 90th quantile is sufficient to represent the high tidal height. Even if mangroves near to the high tidal height are not distinguished from 90th quantile synthetic images, they can be complementally distinguished by a SVM classifier. Thus, that is why we choose 10th and 90th quantile for synthesis.

  1. [Comment]: Is the comparison experiment mentioned in Section 3 results for extraction of all mangroves (including submerged and unsubmerged mangroves) or only for submerged mangroves? The method proposed in this study also adopts SVM for extraction of unsubmerged mangroves, and it would be better to set SMRI and SVM for comparison experiment of submerged mangroves only.

[Response]: Thank you for the comment. In Section 3, the results from SMRI+QS extracted all mangroves including submerged mangroves and unsubmerged mangroves. In fact, SVM classification algorithm can only extract unsubmerged mangroves. For submerged mangrove areas, these areas are visually interpreted as water from Sentinel-2 images. Obviously, SVM should misclassify submerged mangroves as water, and has no ability to extract submerged mangroves. Thus, we did not add additional SMRI+QS and SVM comparison experiment to discuss the effectiveness of distinguishing submerged mangroves.

  1. [Comment]: Mangrove is mainly distributed in the third quarter results part mentioned BeiLunHeKou mangrove reserve (BNNR), and pass mangrove reserve (SNNR) and MAO sea mangrove reserve (MPNR), guangxi readers unfamiliar with these names is not linked to specific areas in the figure, figure 6 best mark out the three regions.

[Response]: We agree with the comment and have marked these three regions in Fig. 6 in the revised manuscript.

  1. [Comment]: The range of spartina alterniflora and mangrove do not completely coincide (FIG. 5e and 5F). FIG. 8 is difficult to explain the impact on mangrove extraction results. It is better to show the initial range of mangrove (before removing spartina alterniflora area) and the final range of mangrove (after removing spartina alterniflora area), and provide quantitative accuracy evaluation results of both.

[Response]: Sorry for the confuse. Actually, some areas not only exist mangroves, but also S. alterniflora, and some areas only have one specie mangroves or S. alterniflora. We only choose one area as a case to describe the flow of the proposed method in Fig.5. Sorry for misunderstanding.

We have added the initial range of mangroves (before removing S. alterniflora) and the final range of mangroves (after removing S. alterniflora) in Fig.8. Quantitative accuracy was evaluated in Section 3 by comparing the proposed method and SVM. We have added quantitative evaluation on result comparison between the initial range of mangroves and the final range of mangroves in Section 4.2 in lines 339-346.

Lines 339-346:

To quantitatively discuss the effect of S. alterniflora on mangrove forest mapping, the initial mangrove forest results (before removing S. alterniflora conducted by the proposed method, called initial SMRI+QS), the final mangrove results (after removing S. alterniflora, called SMRI+QS) and S. alterniflora results were mapped in Fig.9. The area of initial SMRI+QS, SMRI+QS, and S. alterniflora were calculated. The total area of initial SMRI+QS before removing S. alterniflora is 9308.86 ha and the area of SMRI+QS after removing S. alterniflora was 9110.17 ha, respectively. The total area of S. alterniflora is 1088.08 ha. The area before removing S. alterniflora was 198.69 ha larger than that after removing S. alterniflora.

  1. [Comment]: Comparing the discussion section with other mangrove maps, it is difficult to explain the advantages and disadvantages of different products only by area statistics. It is suggested to compare the visual effect and quantitative accuracy evaluation results of the result maps.

[Response]: We are grateful for the suggestion. In the revised manuscript, we collected and used mangrove mapping product from Murray et al. instead of Zhao et al.’s because Zhao et al.’s product was not available in public. We also added the result maps of other products (Fig. 9) and visually discussed the effects of mangrove mapping from the proposed method in Section 4.3. However, different products used different sample points. It is difficult to conduct accuracy evaluation with one set of sample points. Moreover, this research focuses on comparing mangrove forest mapping products, instead of mangrove classification algorithm. Thus, we are sorry that accuracy evaluation was not conducted in the revised manuscript.

  1. [Comment]: The scope of the study area in Figure 1 is not accurate. A large part of the Guangxi border marked in red is not a coastline.

[Response]: We do agree with this comment. We have revised Fig. 1 to make it clear in Line 177.

  1. [Comment]: Figure 2 arrows of spartina alterniflora range mask are not connected properly.

[Response]: Thanks for this comment. We have fixed it in Fig. 2.

  1. [Comment]: It is better to change the scale of the three detail drawings in FIG. 6 and FIG. 8 to the lower left corner of the picture without blocking map elements.

[Response]: We are grateful for the suggestion, and we have revised Fig. 6 and Fig.8 as you suggested in the revised manuscript.

Reviewer 2 Report

  1. Existing mangrove indices include not only 5 species listed in Table 1, but also MVI, NDMI, NIMI, OSCMI, etc.
  2. Why did this study choose 2019-2020 images instead of one year?
  3. The type and number of sample sites in this paper are not clearly expressed: the non-mangrove category should include spartina alterniflora; The number of submerged and unsubmerged mangrove samples in training samples and verification samples is not explained separately, but only the total number of mangrove samples is provided. Distribution map of sample points and quantity table can be provided. 
  4. Why choose 20km as buffer radius?
  5. Quantile synthesis of high tide and low tide images: 10% and 90% in FIG.2 correspond to high tide and low tide respectively, while 10% and 90% in Section 2.3.2 correspond to low tide and high tide respectively. Which one is correct? Also, why 10 percent, 90 percent instead of 5 percent, 95 percent, or whatever?
  6. Is the comparison experiment mentioned in Section 3 results for extraction of all mangroves (including submerged and unsubmerged mangroves) or only for submerged mangroves? The method proposed in this study also adopts SVM for extraction of unsubmerged mangroves, and it would be better to set SMRI and SVM for comparison experiment of submerged mangroves only.
  7. Mangrove is mainly distributed in the third quarter results part mentioned BeiLunHeKou mangrove reserve (BNNR), and pass mangrove reserve (SNNR) and MAO sea mangrove reserve (MPNR), guangxi readers unfamiliar with these names is not linked to specific areas in the figure, figure 6 best mark out the three regions.
  8. The range of spartina alterniflora and mangrove do not completely coincide (FIG. 5e and 5F). FIG. 8 is difficult to explain the impact on mangrove extraction results. It is better to show the initial range of mangrove (before removing spartina alterniflora area) and the final range of mangrove (after removing spartina alterniflora area), and provide quantitative accuracy evaluation results of both.
  9. Comparing the discussion section with other mangrove maps, it is difficult to explain the advantages and disadvantages of different products only by area statistics. It is suggested to compare the visual effect and quantitative accuracy evaluation results of the result maps.
  10. The scope of the study area in Figure 1 is not accurate. A large part of the Guangxi border marked in red is not a coastline.
  11. Figure 2 arrows of spartina alterniflora range mask are not connected properly.
  12. It is better to change the scale of the three detail drawings in FIG. 6 and FIG. 8 to the lower left corner of the picture without blocking map elements.

Author Response

Dear Editor and reviewer,

The reference number of the paper submitted earlier is 1709340, entitled “An improved SMRI-based method for mapping mangrove forests by removing the disturbance of tidal dynamics and S. alterniflora”.

We do appreciate two reviewers as well as you for the guidance and efforts on our manuscript. Given their valuable comments and suggestions, we have made a major revision on our manuscript and all changes made have been marked and addressed in responses to comments. We hope that the revised manuscript is qualified for publication in Remote Sensing.

If you have any questions, please don’t hesitate to contact us.

Yours sincerely,

Qing Xia and Tingting He

On behalf of all the authors

Reviewer #2:

  1. [Comment]: I would suggest not including acronyms in the title.

[Response]: The Reviewer is correct. We have revised SMRI with Submerged Mangrove Recognition Index in the title.

  1. [Comment]: Line 23 – introduce what are S. alterniflora to the reader.

[Response]: We appreciate the Reviewer pointing this out. S. alterniflora was introduced to China in 1979, and rapidly spread along Chinese coastline. Current SMRI failed to separate S. alterniflora from submerged mangrove forests due to similar spectral signatures between them. We have added the introduction of S. alterniflora in Lines 25-28 in the revised manuscript.

  1. [Comment]: Line 25, 26-27 – time series of Sentinel-2 images.

[Response]: Sorry for the mistake. We have modified this expression throughout the manuscript according to the comment.

  1. [Comment]: Line 39 – refer to guidelines to select keywords; select the most influential words.

[Response]: We are grateful for the suggestion. We have revised the keyword referred to guidelines in the revised manuscript.

  1. [Comment]: Line 50 – 60 – Numerous classification methods - I would suggest adding some details about these studies, the classification method, the accuracy obtained, the issue with tides there, and whether they misclassify small mangrove patches under the water?

[Response]: We deeply appreciate the reviewer’s suggestion. We have specified the classification methods in lines 62-68, 75-78 in the revised manuscript.

Lines 62-68:

For example, Conchedda et al. proposed an object-based approach to SPOT XS data to distinguish mangrove forests in Low Casamance, Senegal, and achieved an overall accuracy of 86%. Wang et al. applied a clustering-based neural network classifier to IKONOS data to discriminate mangrove forests in Punta Galeta on the Caribbean coast of Panama. The results showed the overall accuracy of mangroves is 96% and Kappa coefficient is 0.88. Although the researches mentioned above achieved mangrove forest mapping, the effect of tides on mangrove forests remains to be settled.

Lines, 75-78:

For example, Zhang et al. proposed a decision tree-based approach using multi-tidal Landsat 5 data and a Digital Elevation Model to map mangroves. The results were show that considering the effect of tides on mangrove forest mapping can greatly improve the accuracy of mangrove forest mapping.

  1. [Comment]: Line 62 – 72 – it sounds like there are enough mangrove indices; please explain what is missing in each study; Wouldn’t that be an excellent motivation for your study? For example, line 65-66 (MI) —how good MI is for mapping mangroves and how bad it is?

[Response]: We deeply appreciate the reviewer’s suggestion and have pointed out the characteristics of each mangrove index. Please see lines 83-93 in the revised manuscript.

Lines 83-93:

For example, Winarso et al. proposed a mangrove index (MI) using near infrared (NIR) and shortwave infrared (SWIR) bands from Landsat-8 OLI images (Table 1). However, MI was applied in one or two case study and its feasibility in different regions remains to be verified. Jia et al. proposed a mangrove forest index (MFI) using Sentinel-2 data. MFI is effective for distinguishing submerged mangrove forests utilizing red-edge and NIR bands, however, the applicability of MFI is limited due to lack of red-edge band for most remote sensing sensors. Additionally, a combine mangrove recognition index (CMRI) was proposed by Gupta et al. and the presence of vegetation is expressed by NDVI and water information of mangrove forests is expressed by normalized difference water index (NDWI). CMRI has the same problem with MI, namely applying CMRI to more sites to be further studied. However, the mentioned mangrove-specific indices have a common problem. That is, they have not a strong ability to describe tidal variation using a single-tide image.

  1. [Comment]: Line 71 – 73 – does this true for all mangrove species? What are the most affected species? For example, this is not 100% true for the tropical Rhizophora genus; they have long roots to stand up above the tide, and this needs to be addressed very clearly.

[Response]: We are extremely grateful to the reviewer for pointing out this problem. Tides cannot submerge all mangrove species. Tide fluctuation can only submerge seaward low-height mangrove species, such as pioneer species Avicennia marina and low-height Aegiceras corniculatum. For high-height mangrove species, the height difference between the tides is not enough to submerge high-height mangrove species. Our study focuses on distinguish mangroves, regardless of mangrove species. Sorry for the confusion. We have given an explanation at the beginning of this paragraph in lines 80-83 in the revised manuscript.

Lines 80-83:

Tide fluctuation can only submerge seaward low-height mangrove species, such as pioneer species Avicennia marina and low-height Aegiceras corniculatum. For high-height mangrove species, the height difference between the tides is not enough to submerge high-height mangrove species.

  1. [Comment]: Line 105 – 110- what is S. alterniflora? Explain briefly. What are their phonological differences? Do they grow with all mangrove species or have preferences like topography, climate, sea salt level, preferred mangrove species etc?

[Response]: Thanks for this comment. We have added the information on S. alterniflora in lines 122-126 in the revised manuscript.

Lines 122-126:

  1. alterniflora is a perennial salt marsh grass native to North America and introduced to China in 1979 [27]. It has very strong root systems and prefers to grow with pioneer mangrove species in the seaward low-elevation tidal flats. During the past 40 years, S. alterniflora has fast expanded to all over China’s coastal zones and occupied more and more areas.

  1. [Comment]: Line 112 to 123 – what specific objectives of this research should be explained clearly.

[Response]: We deeply appreciate the reviewer’s suggestion. Our study aims to propose a SMRI-based method for rapid and accurate mapping of mangrove forests using time series of Sentinel-2 data and GEE. Specific objectives are listed as below: (1) Apply quantile synthesis to generate high- and low-tide synthetic images and discuss the abilities of synthetic images to distinguish submerged mangrove forests; (2) Utilize phenology-based differences to remove the disturbance of S. alterniflora; (3) Compare the proposed results with other existing mangrove forest products. This information has been added in lines 137-144 in the revised manuscript.

  1. [Comment]: Figure 1 – map should be improved, what is the coordinate system specified there? Make a large-scale study area map and add the location as an inset map.

[Response]: Thanks for the suggestion. We have given a map of China with the study area marked in Fig.1. We also inserted the coordinate system in line 158 in the manuscript.

  1. [Comment]: Line 132 – which species exactly submerged?

[Response]: Thanks for the question. As discussed above in our response to Comment 7, seaward low-height mangrove species, such as pioneer species Avicennia marina and low-height Aegiceras corniculatum, usually are totally or partially submerged. We have provided this information in Lines 80-83.

  1. [Comment]: Line 144 – mangrove forest coverage may not change, but their spectral reflectance may vary— during the rainy season more chlorophyll and dry season less chlorophyll. This will change NDVI values, and I would suggest selecting one season data only.

[Response]: Sorry for the confusion. Maybe there is a misunderstand about NDVI values. NDVI value differs with different seasons. That is why we use NDVI as a key phenological indicator. The average annual NDVI profile was calculate to explore phenological differences between mangrove forests and S. alterniflora. Thus, only one season data is not enough to describe phenological differences. We have given more details in Section 2.3.3.

  1. [Comment]: Line 146 – mNDWI explain. What kind of phonological variations would you expect in these indices raster?

[Response]: We appreciate this comment. Please see the response to comment 15. We have explained why we calculate the time series of NDVI and mNDWI in lines 160-162 in the revised manuscript.

Lines 160-162:

The Modified Normalized Difference Water Index (mNDWI) is used to describe tidal inundation and the Normalized Difference Vegetation Index (NDVI) is a good vegetation indicator for phenological variation. The time series of NDVI and mNDWI were produced to provide indicative information on phenological variations and tidal inundation.

  1. [Comment]: Line 158 – explain quantile synthesis – 25% & 75% or what did you select and why, what would you modify, which parameters, and how would you justify your method not to deviate a lot from reality?

[Response]: We determine 10% and 90% as quantile for low-tide and high-tide synthetic images. There are two reasons as follows. 1) We try to select Sentinel-2 images with less than 20% cloud cover in the image preprocessing stage, however, it is impossible to remove poor-quality images affected by clouds and shadows. Thus, it is not reasonable to take the maximum and minimum at each pixel of mNDWI images as the highest tidal height and lowest tidal height. To remove the disturbance of poor-quality images, we considered quantile synthesis in this study. 2) Based on our visual interpretation, mangrove forests growing in the low-tide synthetic images with quantile less than 10th are difficult to be found. Thus, we selected 10th quantile as the low-tide synthetic image. 90th quantile is sufficient to represent the high tidal height. Even if mangroves near to the high tidal height are not distinguished from 90th quantile synthetic images, they can be complementally distinguished by a SVM classifier. Thus, that is why we choose 10th and 90th quantile for synthesis.

  1. [Comment]: Line 173 – now I can see mNDVI explanation but it should go to line 146. I would suggest you explain the methodology and include the Fig. 2, it might more clearly to the reader.

[Response]: Thanks a lot for your suggestion. We have moved the explanation of mNDVI to line 160 in the revised manuscript. In the original manuscript, we have given an overview and explanation of the proposed method in Section 2.3, and then gave a detail explanation for each part in Section 2.3.1-2.3.4.

  1. [Comment]: Line 191 – how would you address NDVI situation issues, especially in areas having Rhizophora stylus?

[Response]: The NDVI is a common and widely-used remote sensing index, which is an indicator of vegetation coverage and density. The NDVI of a densely vegetated area will tend toward positive values, whereas water or built-up areas will be represented near zero or negative values. In our existing study, we only focus on distinguishing submerged mangrove forests, instead of discriminating submerged mangrove species.

  1. [Comment]: Line 234 – how did you avoid other land cover classes within the study area?

[Response]: We appreciate this comment. To be more clearly to the reviewer, we gave a detail explanation as follows. We input 800 sample points (including submerged mangroves, non-submerged mangroves, S. alterniflora, tidal flats, water, offshore ponds, built-up land) into the SVM classifier for classifying different land cover classes, and then the output results are vector layers of different land cover types (including mangroves and other land cover classes). Each vector layer stored the classification result of each land cover type. We only select a vector layer of mangroves to avoid other land cover classes in the study area.

  1. [Comment]: Line 235 – “to evaluate the performance”

[Response]: Sorry for the typo. We have fixed it in the revised manuscript.

  1. [Comment]: Figure 5 – (1) this is your results; it should go to the results section.

(2) - (a) to (g) make them smaller; legend polygons make smaller; what are the coordinates?

(3) I don’t see an overlap between (e) and (f) in these figures, it seems that there is no S. alterniflora misclassified as mangroves? I would suggest elaborating this in terms of the number of pixels/areas at each class and each method.

[Response]: Thanks for your suggestion.

(1) We only choose one site as a study case to elaborate the flow of the proposed method. We put Fig.5 in section 2.3.4 to make readers more easily and intuitively understand the proposed method;

(2) We have made (a)-(g) and legend smaller. The geographic coordinate datum of the China map is WGS-1984. All images and maps in the study used WGS-1984 geographic coordinate datum. We have explained it in line 159 in section 2.1.

(3) We have recognized that the overlap between S. alterniflora and mangrove forests is not easy to distinguish. To make it clearly, we have added a preliminary mangrove map to (f) and marked the overlap areas in red circle in (e) and (f).

  1. [Comment]: Figure 6 – expect a better map with map elements.

[Response]: Thanks for the suggestion. We have labeled typical study area and National and Provincial Mangrove Nature Reserves in Fig.6 in the revised manuscript.

  1. [Comment]: Line 276 – 278 – what does that mean? Not clear to me.

[Response]: Not sure if we understood this correctly. Do you refer to this sentence “Before distinguishing submerged mangrove forests zone using SMRI, we try to select high-tide and low-tide images, which are as close to the highest and lowest tidal height as possible, respectively”? We are so sorry about this guess. We found that the line number mentioned by reviewer is not consistent with our manuscript. This sentence means that firstly the lowest- and highest-tide images were calculated and obtained by quantile synthesis using time series of mNDWI images, and then the SMRI index was calculated in each pixel. The calculated SMRI result was displayed in the form of a grayscale image.

  1. [Comment]: Line 286 - 288 – this is the first time talking about mangrove forests from SMRI+HL. This should explain the results and Fig. 7 should move to results. No results should be in the discussion. I would suggest making the comparison for the whole study area as well.

[Response]: Sorry for the confusion. We have added the explanation on SMRI+HL in lines 320-325. The results of SMRI+QS and SMRI+HL were also be discussed in lines 326-331 in the revised manuscript.

Lines 320-325:

We selected the highest-tide and lowest-tide images from time series of Sentinel-2 images (2019.01-2020.12), and achieved two mangrove forests mapping results with using quantile synthetic images (SMRI+QS) and without using quantile synthetic images (SMRI+HL)., respectively (Figure 7). SMRI+HL refers to mangrove forest mapping results with two randomly selected high-tide and low-tide images.

Lines 326-331:

From the result of SMRI+QS, the synthetic high-tide and low-tide images can maximum close to the highest and lowest tidal heights, respectively, and then the largest tidal difference can be obtained for more submerged mangrove forests to be detected. From the result of SMRI+HL, mangrove forests under the low tidal height from the selected low-tide image could not be detected, resulting in mangrove area loss.

In this research, Section 3 (result part) focuses on describing the results of the proposed method and one typical classification algorithm (that is SVM classification), and comparing these two results. Section 4 focuses on discussing two advantages of the proposed method, one advantage is applying quantile synthesis, the other advantage is removing the disturbance of S. alterniflora. Section 4.1 mainly focuses on discussing the advantage of using lowest-tide and highest-tide synthetic images and comparing the result derived from using quantile synthesis (SMRI+QS) with the result without using quantile synthesis (SMRI+HL). We put Fig.7 in this section to demonstrate the effectiveness and advantages of using quantile synthesis. Thus, we do not think it is suitable to move Fig.7 to the results. Fig.7 already showed the two comparison results for the whole study area.

  1. [Comment]: Line 302 – Radar

[Response]: Thanks, and we have fixed it.

  1. [Comment]: Line 310 to 312 & 314 to 323 – this is what I was looking for in the beginning, more suitable for the introduction.

[Response]: Thanks for this suggestion. We have moved these sentences to the beginning in lines 122-128 and lines 136-143.

Lines 122-128:

  1. alterniflora is a perennial salt marsh grass native to North America and introduced to China in 1979. It has very strong root systems and prefers to grow with pioneer mangrove species in the seaward low-elevation tidal flats. Because S. alterniflora is characterized by rapid spread and high density, it has become the most serious invasive plant along Chinese coastline. Many S. alterniflora can be found on the fringe of seaward mangrove forests, and had a strong disturbance when mapping mangrove forests.

Lines 136-143:

There are two prominent phenological periods for S. alterniflora including green and senescence periods. These phenological features can be observed across a specific-date Sentinel-2 image. Visually, the color of mangrove forests is much greener than that of S. alterniflora in the senescence period, and this color difference makes them more distinguishable. NDVI values of mangrove forests are much higher than those of S. alterniflora. This phenomenon results in the spectral separation of S. alterniflora. Thus, phenological features derived from time series of Sentinel-2 data were incorporated to eliminate the distribution of S. alterniflora from submerged mangrove forests zone.

  1. [Comment]: Figure 8 – also should go to results. Label inset maps (a) to (c) and describe, add map elements.

[Response]: We deeply appreciate the reviewer’s suggestion. We have inserted (a)-(c) to Fig.8 and revised Fig.8. The reason why we put Fig.8 see the response to comment 22. Section 4.2 mainly focuses on discussing the advantage of removing S. alterniflora using phenological differences and comparing the result derived from removing S. alterniflora with the result without removing S. alterniflora. We put Fig.7 in this section to demonstrate the effectiveness and advantages of removing S. alterniflora using phenological differences. Thus, we do not think it is suitable to move Fig.8 to the results.

  1. [Comment]: Line 341 – what is the meaning of the first sentence, missing something there?

[Response]: Sorry for the mistake. You refer to “Some literature published mangrove forest maps or datasets.”? It means that some researchers have published available mangrove forest mapping results or datasets. We have modified this sentence in line 338.

  1. [Comment]: At the end of the discussion, you should include the effects of different species and tidal inundation and compare areas without submerging mangroves etc.

[Response]: Thanks for the suggestion. Our study only focuses on distinguishing submerged mangrove forests whatever species they are.

In our study, quantile synthesis was used to generate the highest-tidal and lowest-tidal synthetic images. The effect of different tidal inundation has avoided by quantile synthesis. Thus, we do not compare the effect of different tidal inundation. But we discuss the effect of tidal inundation in section 4.1.

The entire Section 3 displayed the result of submerged and non-submerged mangroves and discussed the areas and the advantages of the proposed method.

Round 2

Reviewer 1 Report

Thanks for your responses, however, there are some unanswered comments and a few additional comments to improve your manuscript. Please see the attached document.

Author Response

Dear Reviewer 2,

Thank you for your comments concerning our manuscript entitled “An improved Submerged Mangrove Recognition Index-based method for mapping mangrove forests by removing the disturbance of tidal dynamics and S. alterniflora”. Those comments are valuable and helpful. We have read through comments carefully and have made corrections. All changes made have been marked and addressed in responses to comments.

  1. [Comment]: Line 82 – one or two case studies or you could say a few studies

[Response]: Sorry for the mistake. We have revised it.

  1. [Comment]: What is the relationship between this paragraph and Table 1? Table 1 is a great addition to the manuscript, so that explain its content here, mostly advantages, disadvantages, the ntry to identify knowledge gap.

[Response]: We appreciate this comment. Table 1 lists how these indices mentioned in this paragraph are calculated, and make it easier for readers to understand. Moreover, five additional indices have been added to Table 1 according to Reviewer 1’s comment in the last round of revision.

  1. [Comment]: Line 89 – 93: This is a great addition but need to revise the sentence. This is only a suggestion, it is up to you how to rewrite this information.

[Response]: We are grateful for the suggestion. We have rewritten this information in lines 97-100.

  1. [Comment]: Line 147 – remove “And” from the beginning of the sentence, and make the sentence past tense as you have already completed this step. “…We conducted some experiments.

[Response]: Thanks for the reminding. We have corrected it.

  1. [Comment]: Line 168: should correct as ….……WGS 1984 Geographic Coordinate System.

[Response]: We thank reviewer #2 for pointing out it. We have capitalized the first letter in lines 172-174.

  1. [Comment]: This is my previous comment “Figure 1 – map should be improved, what is the coordinate system specified there? Make a large-scale study area map and add the location as an inset map”. I appreciate your effort but remove the first map, leave 2 & 3. Name them (a) and (b); explain what are (a) and (b) in the figure caption.

[Response]: We deeply appreciate the reviewer’s suggestion. We have modified Figure 1.

  1. [Comment]: Figure 1 – third section, remove “Study Area” in red.

[Response]: Thanks a lot. We have removed it.

  1. [Comment]: Line 180 – check the reference style for web pages.

[Response]: Thanks for this reminding. We have checked it.

  1. [Comment]: Line 187 - 190 – this sounds like a result of my comment below, but it is not clear to me at all. What does this response means, I don’t understand it. I think you better use some references and explain how NDVI, mNDWI relate to phonological variations. What do you expect to see in your images, color differences, numeric differences high/low etc. explain here. The reader cannot interpret you, you have to tell the reader.

[Response]: We apologize for the unclear response in the last round. This round, we got reviewer #2’s idea. We agree completely with reviewer #2 on this, and this has been addressed in lines 196-205. References were also added to explain this issue.

Lines 196-205:

Mangrove forests are periodically inundated by the tide, and the key to describe tidal status is to identify open surface water. In previous studies, the modified normalized difference water index (mNDWI) has been widely used to identify open surface water [43,44,60]. The mNDWI value varies with land cover types. The mNDWI of open surface water area will tend toward positive values, whereas other land cover types will be represented negative values. The Normalized Difference Vegetation Index (NDVI) closely related to green vegetation is a good indicator of vegetation. The NDVI can describe phenological variation for S. alterniflora and mangrove forests, and the NDVI values of mangrove forests are higher than those of S. alterniflora in the senescence period [36].

  1. [Comment]: Figure 2 – use two sides only to show coordinates, this will remove lots of clutter in the map.

[Response]: We appreciate this comment. We have revised Figure 2.

  1. [Comment]: Figure 2 captions, explain clearly.

[Response]: Thanks a lot. We have revised it into “Location map of sample points derived from seven land cover types” to make it clearly.

  1. [Comment]: Line 194 – any suggestion to compensate the inaccuracies caused by the handheld GPS? I would suggest addressing this referring the spatial resolution of images.

[Response]: We thank reviewer #2 for his/her useful suggestion. Positional accuracy of the handheld GPS device used in this study is 5 m, and the spatial-resolution of Sentinel-2 images used is 10 m. Positional accuracy of GPS is accurate enough for Sentinel-2 images. We gave an explanation in line 210 to avoid this concern.

  1. [Comment]: Line 206 – I would suggest a brief explanation of “quantile synthesis” here, that is just the percentages considered within brackets.

[Response]: We appreciate the suggestion. A brief explanation of quantile synthesis is added in lines 227-233.

Lines 227-233:

Quantile synthesis uses quantiles at each pixel of the time-series images to estimate the tidal datum from the pixel. Here, the low and high tidal data were characterized using the 10th and 90th quantile, respectively. The 10th quantile was determined to describe the low tidal datum due to the removal of poor-quality images affected by clouds and shadows. The 90th quantile is sufficient to represent high tidal data based on our visual interpretation, without an assumption about the transit time of satellites.

  1. [Comment]: Line 243 – 244; I am elaborating your below comment. I know what is NDVI but my question is that NDVI saturate where there is healthy vegetation, especially when there are Rhizophora stylus with lots of spongy leaves, means more chlorophyll and more NIR reflectance, then you may end up with wrong results.

[Response]: We appreciate this comment. We completely agree with your viewpoint. Yes, NDVI values increase in the densely vegetated areas, especially when there are Rhizophora stylus with lots of spongy leaves, whereas NDVI values decrease in the sparsely vegetated areas. In this study, we collected 500 mangrove forest samples from different places, different sparsity and different species. Then, we calculated the mean NDVI of 500 samples to generate final NDVI value. A sufficient number of sample points can avoid this concern.

  1. [Comment]: Figure 5 – move to “Results” – it is your results and should not be in the method.

[Response]: Thanks for your suggestion. We have moved it to “Results”.

  1. [Comment]: Line 255 – 260 – move to Results.

[Response]: Thanks a lot. We have moved it to “Results”.

  1. [Comment]: Figure 6 – use smaller font size for (a) to (g) labels (all figures); to make these figures less crowded, add coordinates to left side and top only (??)

[Response]: We appreciate this suggestion. We have revised Figure 6.

  1. [Comment]: 3. Results – this should start with figure 5 – see my comments above.

[Response]: We have fixed it in the revised manuscript.

  1. [Comment]: Line 317 – “Thanks to the accessible Sentinel-2 data.” yes, we should appreciate their service but this is a scientific paper, so we would say “With the increasing availability of Open Data (Sentinel 2) and Open Source programs (GEE), ……………………………….??

[Response]: Thanks for your suggestion. We have fixed it.

  1. [Comment]: This refers to my previous comment (below) and your response; Anyway, your explanation is clearer and it should go to the manuscript rather than that sentence. Sorry about the confusion.

[Response]: We appreciate this comment and added it to the manuscript in lines 407-412.

  1. [Comment]: Again, my previous comment; [Comment]: Line 302 – Radar

[Response]: Sorry for the mistake. We have fixed it.

  1. [Comment]: All figures from “Discussion” should go to “Results”. It doesn’t matter they generated by the study or compare with previous, they are results.

[Response]: We deeply appreciate the reviewer’s suggestion. We have moved all figures from “Discussion” to “Results”.

  1. [Comment]: Line 405: below is my previous comment and your response.

[Response]: We thank reviewer #2 for patiently point out language issues. We have revised this sentence.

  1. [Comment]: Finally, I would recommend a language editing process.

[Response]: Thanks for this suggestion. We did a hard work to polish the language in this manuscript. Additionally, we also have contacted a native-English speaker team to edit the language. Hope it works.
